# Sugar-Sweetened Beverages Intake, Abdominal Obesity, and Inflammation among US Adults without and with Prediabetes—An NHANES Study

**DOI:** 10.3390/ijerph20010681

**Published:** 2022-12-30

**Authors:** Wei-Ting Lin, Yu-Hsiang Kao, Mirandy S. Li, Ting Luo, Hui-Yi Lin, Chien-Hung Lee, David W. Seal, Chih-yang Hu, Lei-Shih Chen, Tung-Sung Tseng

**Affiliations:** 1Social, Behavioral, and Population Sciences, Tulane University School of Public Health and Tropical Medicine, New Orleans, LA 70112, USA; 2Department of Public Health, College of Health Science, Kaohsiung Medical University, Kaohsiung 80708, Taiwan; 3Behavioral and Community Health Sciences Program, School of Public Health, Louisiana State University Health Sciences Center, New Orleans, LA 70112, USA; 4Biostatistics Program, School of Public Health, Louisiana State University Health Sciences Center, New Orleans, LA 70112, USA; 5Environmental and Occupational Health Sciences, Louisiana State University Health Sciences Center, New Orleans, LA 70112, USA; 6Department of Health and Kinesiology, Texas A&M University, College Station, TX 77843, USA

**Keywords:** sugar-sweetened beverages, inflammation, abdominal obesity, prediabetes, NHANES

## Abstract

Excessive sugar-sweetened beverages (SSB) consumption and abdominal obesity have been independently linked to numerous disorders, including diabetes and elevated C-reactive protein (CRP). This study aimed to explore the association between SSB intake, abdominal obesity, and inflammation in normal and prediabetic adults. Sugar intake from SSBs was calculated from 24-h dietary recalls and further classified into non-, medium-, and high-intake. The status of non- and prediabetes was identified based on hemoglobin A1c level. All analyses were performed under a survey module with appropriate sampling weights to control for the complex survey design. A total of 5250 eligible adults without diabetes were selected from the 2007–2010 NHANES. A 1.31-fold increased risk of developing prediabetes was observed in people who consumed high sugar from SSBs when compared to non-SSB consumers. Among individuals with prediabetes, adults who consumed a high amount of sugar from SSB had a 1.57-fold higher risk to increase CRP when compared to non-SSB consumers, even after adjusting for abdominal obesity. Furthermore, the association between the high amount of sugar intake from SSBs and elevated CRP was strengthened by abdominal obesity in prediabetes (*p* for interaction term = 0.030). Our findings highlight that a positive association between sugar intake from SSBs and CRP levels was only observed in US adults with prediabetes. Abdominal obesity may strengthen this effect in prediabetic adults with a high amount of sugar intake from SSBs.

## 1. Introduction

C-reactive protein (CRP) is an acute-phase reactant primarily secreted by macrophages and fat cells. An elevated level of CRP in plasma has been linked to inflammation, injury, or bacterial infection [1]. Prior studies have shown that low-grade inflammation is associated with the development of diabetes, cardiovascular disease, and cancer [2,3]. Elevated CRP has been found to be strongly related to increased diabetes development due to the biological mechanism of low-grade chronic inflammation in glucose metabolism disorders [4,5]. Some recent studies demonstrated that high sugar sweetened beverage (SSB) consumption promotes inflammation [6,7]. One recent study suggested that reducing SSB intake is beneficial for glycemic and waist circumference (WC) control, and a lower risk of developing diabetes-related adverse health outcomes in diabetes [8]. However, no study explored the association between SSB intake and the risk of elevated CRP in prediabetes.

Hemoglobin A1c (HbA1c) is a widely used tool to routinely assess and monitor long-term glycemic control in both diabetes and prediabetes [9]. An HbA1c level between 5.7% and 6.4% is a precursor to diabetes diagnosis for prediabetes [9,10]. According to the National Diabetes Report, 33.9% (84.1 million people) of US adults with prediabetes were reported in 2015 [11]. Around 1.5 million Americans are newly diagnosed with diabetes each year [12]. Findings from the National Health and Nutritional Examination Survey (NHANES) show an increasing prevalence of prediabetes, from 10.02% in 1988–1994 to 18.5% in 2011–2012, was observed. However, most individuals are undiagnosed and unaware of their prediabetes status [13,14].

WC has been suggested as a better, simpler, and less expensive surrogate anthropometric measurement than body mass index (BMI) for abdominal obesity identification [15]. More than 50% of US adults had abdominal obesity, based on the 2013–2014 NHANES report [16]. Abdominal obesity has been linked to an increased risk of diabetes development in a prior study [17]. Based on the biological mechanism in the liver, abnormal adiposity is also a risk of low-grade inflammation [3,18]. But limited studies examined how abdominal obesity influences the association between SSB consumption and CRP levels in populations with and without prediabetes. Therefore, the purpose of this study is twofold: first, to investigate the different associations of sugar intake from SSBs and elevated CRP among US adults without and with prediabetes; and second, to evaluate the interplay effect between sugar intake from SSBs and abdominal obesity on elevated inflammation in individuals without and with prediabetes.

## 2. Materials and Methods

### 2.1. Study Design and Population

We used the National Health and Nutrition Examination Survey (NHANES), a nationally representative survey of the US population, which is conducted every two years by the Centers for Disease Control and Prevention’s National Center for Health Statistics (NCHS). In this study, 2007–2010 NHANES data were combined and examined in this study. Survey sampling weights, sociodemographic and lifestyle factors, personal medical conditions, dietary pattern, physical examination, and laboratory data were extracted from NHANES surveys.

Two steps to identify eligible subjects were performed in this study. We first excluded subjects who self-reported with diabetes based on the diabetes-related question; “Have you been told by doctor or health professional that you have diabetes?”. Second, individuals who had HbA1c levels greater than 6.5% were also excluded from our study subjects based on the suggested criteria [10]. All detailed procedures, including survey methods, design, operations, and analytic guidelines are described elsewhere [19] and were approved by the Institutional Review Board for the Ethics Review Board of the National Center for Health Statistics before conducting surveys. Written informed consent was obtained from each participant [20].

### 2.2. Covariates

Data on age, sex, race, poverty income ratio (PIR), cigarette smoking status, alcohol use, physical activity, and medical condition were obtained via self-reported questionnaires. Nonsmoker, former smoker, or current smoker was classified based on two questions: (1) “Have you smoked at least 100 cigarettes in your lifetime?” and (2) “Do you now smoke cigarettes?”. According to the alcohol use questionnaire, participants who consumed less than 12 drinks/lifetime, ≥12 drinks/past year and ≤5 drinks/day), ≥12 drinks/past year, or >5 drinks/day were defined as nonalcohol drinkers, light, and heavy alcohol drinkers, respectively.

The physical activity questionnaire (PAQ) was changed and based on the Global Physical Activity Questionnaire (GPAQ) since NHANES 2007–2008. The PAQ is used to assess how many days per week and how much time per day was spent doing different intensities of daily activities, leisure time activities, and sedentary activities in a week. Participants who spent at least 150 min doing moderate-intensity activities or at least 75 min doing vigorous-intensity activities during leisure time per week were classified as having adequate physical activity, according to WHO recommendations for adult physical activity [21].

Types of personal health conditions and medical history, such as asthma, chronic bronchitis, emphysema, heart diseases, gout, arthritis, stroke, coronary heart disease, angina, heart failure, heart attack, anemia, any liver conditions, and multiple types of cancer were obtained from a self-reported interview. Subjects were then classified into a chronic disease group.

### 2.3. Dietary Information

A face to face interview was conducted for each participant by a dietary interviewer to recode nutritional information on two separate days using 24-h dietary recall interviews and furtherly calculate into daily nutrients consumption, such as daily total energy intake and daily total sugar intake [22]. The average total energy intake and total sugar intake from two separate days were calculated in this study. Types and amounts of food and beverage consumption were estimated by diverse portion size tools. According to a prior study, any beverage with added sugars, such as soft drinks, sports drinks, energy drinks, fruit-flavored sweetened drinks, artificial fruit juices, sweetened teas and coffees, and other sweetened drinks (such as traditionally sweetened drinks), were defined as SSBs in this study [23]. We identified and summarized the types and amounts of sugar intake from SSBs based on United States Department of Agriculture (USDA) codes and the Nutrient Database for Dietary Studies in the Day-1 and Day-2 individual foods sections of the interview. Detailed examination protocols and procedures are described on the NHANES website [22]. Based on the average of 40 g of sugar in one can of soda (12 ounces), we categorized sugar intake from SSBs into three groups: non-SSB consumers, medium (<40 g/day), and high (≥41 g/day).

### 2.4. Adiposity and Biochemical Examinations

Anthropometric measurements and biospecimens were collected in each mobile examination center (MEC). Waist circumference (WC) was measured to the nearest 0.1 cm; by trained staff using standard protocols with steel tape measures [19]. We defined men with WC > 102 cm (40 in) and women with WC > 88 cm (35 in) as having abdominal obesity [24].

Blood specimens were obtained, stored at −20 °C, and shipped to the Fairview Medical Center Laboratory at the University of Minnesota for analyses. Detailed analytic processes of blood specimens are described in the description of laboratory methodology section [25]. Glycohemoglobin measurements were performed using the A1c G7 HPLC Glycohemoglobin Analyzer. According to new clinical recommendations, NHANES participants with levels of HbA1c ranged from 5.7 to 6.4% were defined as having prediabetes [10]. Study subjects who had a CRP concentration of more than 10 mg/L were also excluded to minimize the influence of acute infection [26]. A cut-off value of elevated CRP was ≥3 mg/L in this study based on the suggestion in prior findings [26].

### 2.5. Statistical Analysis

Appropriate sampling weights, strata, and primary sampling units were used to adjust for the complex sampling design of NHANES [19]. All statistical analyses were performed under survey modules in STATA v15 (StataCorp., College Station, TX, USA). All *p* values < 0.05 were considered significant. For continuous and categorical variables, means, standard errors, and proportions were presented in the descriptive statistical analysis. The independent samples t-test and the chi-square test were performed for continuous and categorical variables, respectively, to evaluate the differences in demographic lifestyle pattern, personal medical condition, dietary factors, and examination outcomes between US adults with low and elevated CRP. To consider individual factors in relation to CRP levels, factors in Table 1 were further adjusted in Table 2. To assess the relationship between sugar intake from SSBs, HbA1c, and risk of elevated CRP, multivariable logistic regression models were performed and presented by an adjusted odds ratio (aOR) and 95% confidence interval (CI). Multivariable logistic regression models with interaction terms were assessed to examine the interplaying effect of abdominal obesity, HbA1c status, and sugar intake from SSBs on elevated CRP. Stratified analysis was further performed to evaluate the different effects of SSBs, abdominal obesity, and risk of elevated CRP between individuals without and with prediabetes. Potential confounders include age, sex, race, PIR, cigarette smoking, alcohol drinking, physical activity, personal medical condition, total daily energy intake, and total sugar intake from diet were considered in this study.

## 3. Results

A total of 5020 adults above 20 years of age without diabetes who had completed at least one day 24-h dietary recall interview, body adiposity measurement, HbA1c level, and CRP examination were selected for final analysis. After controlling for appropriate sampling weights, the distributions for age, sex, race, PIR, cigarette smoking, alcohol consumption, physical activity, and personal disease history between low and elevated CRP were shown in Table 1. A higher prevalence of elevated CRP was found in study subjects who were older (47.9 ± 0.5 years), female (61.9%), non-Hispanic black (15.4%), current smokers (22.8%), low physical activity (70.5%), and with personal medical conditions (47.2%) (all *p’s* ≤ 0.019).

After adjusting for demographic factors, lifestyle patterns, and personal medical conditions, the amounts of sugar intake from SSBs significantly varied between low and elevated CRP groups. In the elevated CRP group, 38.3% of subjects consumed ≥41 g of sugar from SSBs (*p* = 0.020) per day. Additionally, 73.6% of adults with abdominal obesity had elevated CRP (*p* < 0.001). Adults who had prediabetes possessed a higher prevalence of elevated CRP than adults with normal HbA1c (*p* < 0.001) (Table 2).

Table 3 demonstrates the effect of sugar intake from SSBs on elevated CRP between US adults without and with prediabetes after considering for potential confounders. A high amount of sugar intake from SSBs positively associated with a 1.31 times higher risk of developing prediabetes (95%CI = 1.03–1.68) was observed when controlling for abdominal obesity and individual covariates (data was not shown in the table). A significantly increased risk of elevated CRP was found in populations with prediabetes who consumed a higher amount of sugar from SSBs (≥41 g/day) when compared to non-SSB consumers after controlling for abdominal obesity (aOR = 1.57, 95%CI = 1.05–2.34) (model 2). Among individuals with normal HbA1c levels, we did not observe any significant associations between sugar intake from SSBs and a higher risk of having elevated CRP in both model one and model two.

In order to understand whether abdominal obesity modifies the association between sugar intake from SSBs and CRP levels, the stratified analyses were performed. A 1.69- and 2.66-fold increased risk of developing elevated CRP was observed in populations with prediabetes and abdominal obesity who consumed medium and heavy amounts of sugar intake from SSBs, respectively (all *p* ≤ 0.015). The effect modification was detected in adults with prediabetes who were abdominally obese and consumed heavy amounts of sugar from SSBs (*p* for the interaction term is 0.030) (Figure 1).

## 4. Discussion

In this study, we evaluated the difference in sugar intake from SSBs and abdominal obesity associated with inflammation status between US adults without and with prediabetes. The main findings of this study indicated that high consumption of sugar from SSBs was positively associated with elevated CRP levels among individuals with prediabetes. An effect modification of abdominal obesity on this association was also explored in individuals with prediabetes.

Excessive daily energy and sugar consumption from SSBs associated with obesity, metabolic disorders, and low-grade chronic inflammation were reported [27]. A few studies also proposed that higher inflammatory biomarkers were observed in individuals who consumed a high frequency/amount of SSBs [6,27]. Consistently, we found the highest percentage of elevated CRP levels was observed in subjects who consumed high amounts of sugar from SSBs when controlling for demographic factors, lifestyle patterns, and personal medical conditions. However, we did not observe any differences in total energy intake and total sugar intake from diet between populations with low and elevated CRP. Unlike total energy and total sugar intake, SSB consumption is a liquid form of high added sugar content, zero nutrient, and low satiety that may promote visceral adiposity accumulation and further result in low-grade inflammation [7,28,29] Body adiposity is also a strong risk factor for the induction of low-grade chronic inflammation [3,28]. WC, a commonly used simple way to measure and represent abdominal obesity, is a better predictor of type two diabetes than BMI [15,17]. Abdominal obesity, especially visceral fat tissue (VFT), is realized to produce adipokines. Adipokines are cytokines secreted by adipose tissue that have been strongly linked to a higher level of inflammatory biomarkers [3]. According to a possible biological mechanism, inflammatory cytokines, such as CRP, IL-6, and TNF-α, may have key roles in explaining the link between CRP and adiposity [29,30]. Additionally, some findings showed that a stronger link has been found between cytokines and abdominal adiposity than BMI [31,32]. Researchers indicated that obesity should play a vital role in the increased secretion of cytokines such as IL-1, IL-6, IL-8, TNF-α, and CRP, and further lead to chronic low-grade inflammation [29,30]. In this study, 73.6% of US adults with abdominal obesity had elevated CRP levels and were evaluated after adjustment for demographic factors, substance use, physical activity, and personal medical conditions. Therefore, researchers suggested that body adiposity should be considered while investigating the association between SSB intake and inflammation [6,27]. In the present study, a significantly increased risk of having elevated inflammation was evaluated in individuals with prediabetes who consumed a high amount of sugar from SSBs when compared to non-SSB consumers, even after additionally adjusting for WC.

The Framingham Offspring cohort study showed that high SSB consumers had a 46% higher risk of developing prediabetes than non-SSB consumers [33]. In this study, 30.6% of US adults with prediabetes was observed to have elevated CRP when adjusted for demographic factors, substance use, physical activity, and personal medical conditions. Based on the biological pathway, increased CRP level is associated with irregularities of β-cell function and insulin secretion [5,34]. Insulin resistance is a major cause of type two diabetes and prediabetes [33]. Most evidence suggested that insulin resistance may be caused by excess sugar intake itself or caused by body adiposity accumulation [33]. The possible biological pathway of hepatic insulin resistance may be explained by increased free fatty acids (FFAs) that result from the accumulation of fat in the liver and adipose tissues after excess sugar consumption. Insulin resistance is induced by the inhibition of insulin-stimulated glucose metabolism, which is caused by increased FFAs from expended adipose tissues [35]. The current study further explored how abdominal obesity modifies the association between sugar intake from SSBs and elevated CRP among non- and prediabetes US adults. Among the abdominally obese population, an increased risk of elevated CRP was estimated in US adults with prediabetes who were consumers of a medium or high amount of sugar from SSBs, when compared to non-SSB consumers with normal HbA1C.

High intake of SSBs has been considered to play a vital role in inflammation, type two diabetes, and adiposity [19,28,35] due to excessive added sugar intake and zero nutrition. For people with diabetes, reducing SSB intake has been suggested for glycemic and WC control in order to prevent diabetes-related adverse health outcomes [8]. Our findings demonstrated that abdominal obesity may magnify the association between high sugar intake from SSBs and elevated CRP, especially in individuals with prediabetes. The greatest risk of elevated CRP was detected in adults with prediabetes who were abdominally obese and consumed a high amount of sugar from SSBs. Based on biological mechanisms, the positive relationship between VAT and insulin resistance may be illustrated by increased levels of FFAs and inflammatory cytokines [30]. In the current study, abdominal obesity may be not only a confounder but a modifier of the association between heavy sugar intake from SSBs and elevated inflammation in individuals with prediabetes. Our findings support that limiting SSB consumption is a simple way to decrease total daily added sugar intake, and is additionally beneficial for abdominal obesity control, inflammation reduction, and prevention of obesity-related disorders and chronic diseases.

This study had several strengths. First, we focused on exploring the effect of sugar intake from SSBs and abdominal obesity on elevated inflammation levels in US adults, especially those who were unaware of prediabetes. Second, the amount of sugar intake from each type of sweetened drink with added sugar was carefully identified and estimated using suggested food codes from the USDA Food and Nutrient Database for Dietary Studies. Third, total energy intake and sugar intake from other sources were calculated and controlled when we explored the effect of sugar intake from SSBs on inflammation. Finally, our findings should be representative of the nationwide US population because NHANES data was used in this study.

This study also had a few limitations. First, physical activity data was measured by a different questionnaire before NHANES 2007–2008. The same procedure and method was used to analyze glycohemoglobin from NHANES 2007–2008. Furthermore, CRP data was available until NHANES 2009–2010. To consider the consistent measurements of data collection and available data, NHANES 2007–2010 is the most recent data that can be combined and performed to explore our study purpose. Second, the nature of the cross-sectional survey data, which cannot infer a causal relationship between sugar intake from SSBs, abdominal obesity, and inflammation status. Second, in order to achieve an adequate sample size, HbA1c was the only biomarker that was available for all participants to report glucose metabolism, which may underestimate the actual number of individuals with prediabetes [36]. Third, recall bias and misreporting of dietary intake may happen in this study due to 24-h recall interviews. Additionally, personal disease information might have been underestimated, because participants may have been unaware of diseases or chose not to report personal medical conditions.

## 5. Conclusions

These findings emphasize the association between the high consumption of sugar from SSBs and elevated CRP levels in adults with prediabetes. Furthermore, abdominal obesity may modify this effect in individuals with prediabetes who consumed medium and heavy amounts of sugar from SSBs. Thus, reducing dietary SSB consumption may be a simple way to improve personal health and lower the risk of obesity-related adverse health outcomes, especially in individuals who had prediabetes with abdominal obesity.

## Figures and Tables

**Figure 1 ijerph-20-00681-f001:**
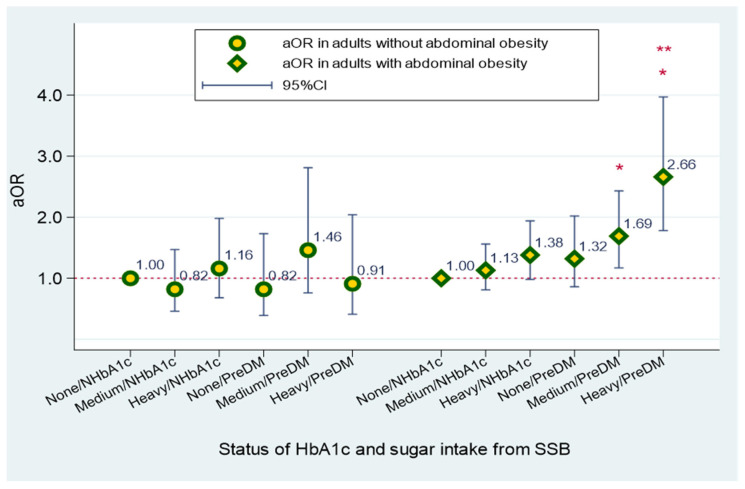
Effect modification of abdominal obesity on sugar intake from SSB and elevated CRP among non- and prediabetes US adults. Abbreviation: SSB, sugar-sweetened beverages; CRP, C-reactive protein; None, non-SSB consumers; Medium, medium amount of sugar intake from SSB (<40 g); Heavy, heavy amount of sugar intake from SSB (≥41 g); NHbA1c, Normal HbA1c (<5.7%); PreDM, prediabetes (HbA1c ≥ 5.7%). * *p* value < 0.05. ** P for interaction term = 0.030. Legends: P for interaction terms was detected in high amount of sugar intake from SSB subjects with abdominal obesity (P for interaction term = 0.030). Compared to non-SSB consumers, a 1.69- and 2.66-fold increased risk of elevated CRP was estimated in prediabetic individuals with abdominal obesity who consumed medium and high amount of sugar from SSB.

**Table 1 ijerph-20-00681-t001:** Distribution of demographic, substance use, lifestyle pattern, disease history, and examinations between the level of C-reactive protein <3 and ≥3 mg/L.

	C-Reactive Protein (mg/L)		
	<3	≥3	*p* Value
**Row population ^a^**	3780	1470	
**Survey-weighted ^b^**			
**Demographic factor**			
Age (year), mean ± se	45.6 ± 0.4	47.9 ± 0.5	<0.001
Gender			<0.001
male	51.2%	38.1%	
female	48.8%	61.9%	
Race			<0.001
non-Hispanic white	89.5%	84.6%	
non-Hispanic black	10.5%	15.4%	
PIR			<0.001
below poverty	10.1%	14.9%	
1–2.9	30.4%	35.9%	
≥3	59.5%	49.2%	
**Substance use**			
Cigarette smoking			0.019
non-smokers	54.5%	50.8%	
former smokers	27.8%	26.5%	
current smokers	17.7%	22.8%	
Alcohol drinking			0.802
non/light drinkers	92.7%	92.4%	
heavy drinkers	7.3%	7.6%	
**Physical activity**			<0.001
low	58.0%	70.5%	
high	42.0%	29.5%	
**Personal medical conditions ^c^**			<0.001
no	61.0%	52.8%	
yes	39.0%	47.2%	

^a^ The raw number was not adjusted for sample survey design. ^b^ The results were adjusted for sampling weight. ^c^ Types of disease included asthma, chronic bronchitis, emphysema, heart diseases, gout, arthritis, stroke, coronary heart disease, angina, heart failure, heart attack, anemia, any liver conditions, thyroid problem, and multiple types of cancer.

**Table 2 ijerph-20-00681-t002:** Characteristics of dietary pattern, sugar intake, adiposity index, and hemoglobin A1c levels between C-reactive protein <3 and ≥3 mg/L.

	C-Reactive Protein (mg/L)		
	<3	≥3	*p* Value
**Row population ^a^**	3780	1470	
**Survey-weighted ^b^**			
**Dietary pattern**			
Total energy intake (Kcal/day), mean ± se	2180 ± 22	2046 ± 24	0.682
Total sugar intake from diet (gram/day), mean ± se	80 ± 1.3	75 ± 2.0	0.224
**SSB-related factor**			
Sugar intake from SSB (gram/day)			0.020
non-intake	28.3%	25.9%	
1–40 g	38.1%	35.8%	
≥41 g	33.6%	38.3%	
**Abdominal obesity**			<0.001
no	57.8%	26.4%	
yes	42.2%	73.6%	
**HbA1c status**			<0.001
normal (HbA1c < 5.7%)	80.4%	69.5%	
pre-diabetes (HbA1c = 5.7–6.4%)	19.6%	30.6%	

^a^ The raw number was not adjusted for sample survey design. ^b^ The results were adjusted for sampling weight. *p* values were adjusted for demographic factors, substance use, physical activity, and personal medical conditions. Abbreviation, SSB, sugar-sweetened beverage; HbA1c, hemoglobin A1c.

**Table 3 ijerph-20-00681-t003:** The adjusted odds ratios (aOR) for elevated C-reactive protein (≥3 mg/L) associated with sugar intake from sugar-sweetened beverage (SSB) in non-diabetes subjects with different status of hemoglobin A1c.

	CRP, ≥3 mg/L vs. <3 mg/L							
	Model 1 ^a^				Model 2 ^b^ Abdominal Obesity-Adjusted Model			
	HbA1c				HbA1c			
	<5.7%		5.7–6.4%		<5.7%		5.7–6.4%	
	aOR	(95%CI)	aOR	(95%CI)	aOR	(95%CI)	aOR	(95%CI)
**Sugar intake from SSB (gram/day)**								
non-intake	1		1		1		1	
1–40 g	0.95	(0.73, 1.25)	1.30	(0.91, 1.87)	1.00	(0.76, 1.32)	1.34	(0.89, 2.03)
≥41 g	1.23	(0.88, 1.71)	1.50	(0.99, 2.26)	1.33	(0.97, 1.83)	1.57	(1.05, 2.34)

^a^ Model 1 was adjusted for covariates, including age, gender, race, PIR, alcohol, smoking, medical condition status, total energy intake, sugar intake from diet, and physical activity. ^b^ Models were adjusted for covariates in Model 1 and abdominal obesity. The cutoff value for the diagnosis of abdominal obesity was waist circumference >102 cm (40 in) in men and >88 cm (35 in) in women. Abbreviation, CRP, C-reactive protein; HbA1c, hemoglobin A1c; SSB, sugar-sweetened beverage.

## Data Availability

Not applicable.

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
