# Peer review of "Sugar-Sweetened Beverages Intake, Abdominal Obesity, and Inflammation among US Adults without and with Prediabetes—An NHANES Study"

_ijerph, 2022, doi:10.3390/ijerph20010681_

Round 1
Author Response
Thanks for the reviewer's comments. Please see the attachment.

Reviewer 2 Report
I carefully read and checked the article. First of all, I would like to state that I did not see any typos. The abstract part summarizes the study well and the important results of the study are given. In the Introducton section, general information is related to each other, a good introduction to the subject is made and the objectives are mentioned. The design of the study is very good, the study design, the inclusion criteria, the exclusion criteria are clear at the study, the data recorded from the system are explained in detail. Statistical methods are explained clearly and data are explained with correct methods. The findings section is very detailed, the tables are very explanatory and clearly prepared. After mentioning the main results in the discussion part, a good discussion was prepared by comparing the findings with the literature data. All the important findings are discussed very clearly. The difficulties and limitations of the study were also mentioned, and the main result obtained from the study was clearly explained in the conclusion part. I read this article with pleasure, good information was given, good results were obtained.
The main argument of the research; excessive consumption of foods containing excessive amounts of sugar triggers abdominal obesity and this excessive intake is associated with obesity, inflammation, and CRP levels. In the literature, there are studies on this subject, especially on diabetes and metabolic syndrome, but I saw that there are not enough studies in non-diabetes and pre-diabetes patients. For this reason, I think the subject is original enough. The article is well written. The writing is very clear, understandable, I had no trouble reading it. The English writing is very fluent and legible. The results are pretty consistent. The results are explained very clearly. The main question asked in the article is clearly answered. For this reason, I think it is acceptable in its current form.Author Response
Many thanks for the reviewer’s time and comments for this article.

Reviewer 3 Report
Dear Authors,
Overall, a well written and presented paper that addresses one of the very important public health issues. Please find below several suggestions and comments
Abstract
Line 20 - add 'including' after ...' multiple disorders'
Line 22 - remove 'interviews' and replace it with recalls
Line 28 - add 'who' after adults
Lines 30-32 Furthermore, abdominal obesity significantly moderated the association between sugar intake from SSB and elevated CRP in prediabetes (p for interaction term = 0.030). - this is not clear
Lines 32 - 33 - Our findings highlight that sugar intake from SSB is positively associated with CRP levels in US adults with prediabetes. It would be worth making it clear if the association was only observed in prediabetic patients.
Introduction:
Line 38-39 - ...has been shown to increase 1,000-fold in response to low grade chronic inflammation... - it is rather one of the markers of inflammation including low grade inflammation, not a response, but yes it is produced in response to injury
Line 44 - add metabolism after glucose
Line 66 - add who before 'consumed'
Materials and Methods
Statistical Analysis - it is not clear what statistical tests have been used e.g Table 1 - demographic characteristic of the population; only p-values mentioned; please add a description of statistical test to the methods section and under the table as different tests were used for continuous and categorical variables similar table 2 not clear what tests were used
Results
Line 216 - no figure included
Author Response
Many thanks for the reviewer’s time and comments for this article. Please see the attachment.

Round 2
Reviewer 1 Report
Dear Authors,
I believe the manuscript can be published as the revised version.
I have no more comments